



# Genesis dynamics of the Angola-Benguela Frontal Zone

**Shunya Koseki[1], Hervé Giordani[2],** and **Katerina Goubanova[3,4],**

1. Geophysical Institute, University of Bergen, Bergen/Bjerknes Centre for Climate Research, Norway

2. Centre National de Recherches Météologiques, MÉTÉO-France, Toulouse, France

3. Centro de Estudios Avanzados en Zonas Áridas, La Serena, Chile

4. CECI/CERFACS-CNRS, Toulouse, France

Correspondence to Shunya Koseki

Email: Shunya.Koseki@gfi.uib.no

Address: Geophysical Institute, University of Bergen, Postboks 7803, 5020, Bergen, Norway

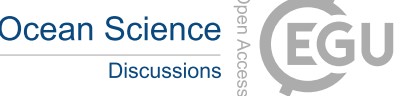

# 1 Abstract

A diagnostic analysis of the climatological annual mean and seasonal cycle of the
Angola Benguela Frontal Zone (ABFZ) is performed applying an ocean frontogenesis
function (OFGF) to the ocean mixing layer (OML). The OFGF reveals that
meridional confluence and the vertical tilting terms are the most dominant
contributors to the frontogenesis of the ABFZ. The ABFZ shows a well-pronounced
semi-annual cycle with two maximum (minimum) peaks in April-May and
November-December (February-March and July-August). The development of the
two maxima of frontogenesis is due to two different physical processes: enhanced
tilting form March to April and the meridional confluence from September to
October, respectively. The strong meridional confluence in September-October is
closely related to the seasonal southward intrusion of tropical warm water to the
ABFZ that seems to be associated with the development of the Angola Dome
northwestern of the ABFZ. The strong tilting effect from March to April is attributed
to the meridional gradient of vertical velocities whose effect is amplified in this
period due to increasing stratification and shallow OML depth. The proposed OFGF
can be viewed as a tool to diagnose the performance of CGCMs that generally fail in
simulating realistically the position of the ABFZ, which leads to huge warm biases in
the southeastern Atlantic.



# 1. Introduction

The Angola-Benguela Frontal Zone (ABFZ, see Fig. 1), situated off the coast
of Angola/Namibia, is a key oceanic feature in the southeastern Atlantic Ocean. The
ABFZ separates the warm sea water of the Angola Current (e.g., Kopte et al., 2017)
from the cold sea water associated with the Benguela Current/upwelling system (e.g.,
Mohrholz et al., 2004; Colberg and Reason, 2006; Veitch et al., 2006; Colberg and
Resason, 2007; Fennel et al., 2012; Goubanova et al., 2013; Junker et al., 2015;
Junker et al., 2017; Vizy et al., 2018). The ABFZ is characterized by smaller spatial
extent and weaker SST gradient compared to the major oceanic fronts generated by
the western boundary currents (Fig. 1). However, due to its near coastal location, the
ABFZ plays important roles for the southern African continent, strongly impacting
local marine ecosystem (e.g., Auel and Verheye, 2007; Chavez and Messié, 2009) and
regional climate over the southern African Continent (Hirst and Hastenrath, 1983;
Rouault et al. 2003; Hansingo and Reason, 2009; Manhique et al., 2015). In
particular, the main model of interannual variability of SST in the ABFZ, so-called
Benguela Niño/Niña (e.g., Florenchie et al., 2003; Rouault et al., 2017), influences the
local rainfall along the southwestern African coast of Angola and Namibia (Rouault et
al., 2003; Lutz et al., 2015) and tends to have a remote impact on rainfall activity over
the southeastern African continent (e.g., Manhique et al., 2015).
The ABFZ region also poses one of the major challenges for the global climate
modeling community. Most coupled general circulation models (CGCMs) exhibit a
huge warm SST bias in the ABFZ (e.g., Zuidema et al., 2016) and fail to reproduce
the realistic SST, its seasonal cycle and the right location of the ABFZ (e.g., Koseki et
al., 2017). While Colberg and Reason (2006) and Giordani et al. (2011) concluded
that the position of the ABFZ is controlled to a large extent by the local wind stress



curl, Koseki et al. (2017) elucidated that the local wind stress curl bias in GCM
contributes partly to the warm SST bias in the ABFZ via erroneous intrusion of
tropical warm water, which is induced by the negative wind stress curl and enhanced
Angola Current. In order to reduce this kind of model biases, one need to understand
the processes of generation of the ABFZ.
Previous studies have focused mainly on SST variability at seasonal and
interannual scales in the ABFZ and its impacts on regional climate are well-studied
(e.g., Rouault et al., 2003; Lutz et al., 2015). To our knowledge, there are no works
insightfully investigating dynamical and thermodynamical processes which generate
and maintain the ABFZ and its seasonal cycle. A dynamical diagnosis for the SST
front in the north of the Atlantic Cold Tongue (e.g., Hasternrath and Lamb, 1978;
Giordani et al., 2013) was proposed by Giordani and Caniaux (2014, hereafter
referred as GC2014). This frontogenetic function is, in general, adapted to explore
sources of frontogenesis of atmospheric synoptic-scale cyclones at the extratropics (e.
g., Keyser et al., 1988; Giordani and Caniaux, 2001). Using a frontogenetic function
GC2014 showed  clearly that the convergence associated with the northern South
Equatorial Current and Guinea Current forces the SST-front intensity (frontogenetic
effect) and mixed-layer turbulent flux destroys the SST-front (frontolytic effect) in
climatology. Fundamentally, the frontogenetic function consists of three mechanical
terms (confluence, shear and tilting) and two thermodynamical terms (diabatic heating
and vertical mixing). Around the ABFZ, all these terms can be considered as
contributors to the frontogenesis due to: (1) two opposite outstanding ocean current
systems, the Angola and Benguela currents (confluence and shear). (2) strong coastal
upwelling (tilting) associated with Benguela current; (3) one of the largest and more
persistent stratocumulus cloud deck in the world (diabatic heating related to radiation)



associated with the cold SST and subsidence due to St. Helena Anticyclone (e.g.,
Klein and Hartmann, 1993; Pfeifroth et al., 2012). So far, the relative roles of these
different processes in the frontogenesis of the ABFZ still need to be investigated.
In this study, following the fundamental philosophy of GC2014, we attempt to
understand the mechanisms responsible for the ABFZ development at seasonal scale
based on a first-order estimation. We propose an ocean frontogenetic function in a
different way from GC2014 (this study focuses on the ocean-mixed layer mean front).
The structure of the remainder of this paper is as follows: Section 2 gives details of
data set used in this study. In section 3, we derive the ocean frontogenetic function.
Section 4 provides a description of the climatological state around the ABFZ. In
section 5, we apply our diagnostic methodology to the ABFZ and determine the main
terms of the frontogenetic function controlling its annual cycle. The associated
processes are discussed in section 6. Finally we summarize and put some concluding
remarks in section 7.

**2. Data**
For an overview of SST and its meridional gradient in the ABFZ and
evaluation of reanalysis data, we employ the Optimum Interpolated Sea Surface
Temperature (OISST, Reynolds et al., 2002) released by National Ocean and
Atmosphere Association (NOAA) that has a quarter degree of horizontal resolution
and daily temporal resolution from 1982 to 2010. For the 3-dimensional diagnostic
analysis of the ABFZ, we utilize 1-hour forecast data of Climate Forecast System
Reanalysis (CFSR, Saha et al., 2010) developed by the National Centers for
Environmental Prediction (NCEP). The ocean component of this system is based on



MOM version 4p0d (Griffies et al., 2004). This system provides 6-hourly data with a
0.5 degree horizontal resolution and 70 vertical layers for ocean. In this paper we will
analyze daily-means. Data of sea water potential temperature (hereafter, referred to as
temperature) is used for the analysis because sea water temperature and sea water
potential temperature are almost identical in the upper ocean layers.

### 3. Ocean Frontogenesis Function


The ocean frontogenetic function (OFGF) is defined and applied to the ocean
mixing layer (OML) in order to propose a dynamical diagnosis of the
maintenance/generating process of the ABFZ. Following GC2014, we use the OFGF
as a tool to unravel the Langrangian (pure) sources of the oceanic front. While there
are plentiful numbers of literature investigating the ocean front dynamics (e.g.,
Dinniman and Rienecker, 1999), the concept of this OFGF has been hardly referred.
The Lagrangian frontogenesis function, $F$, is defined as,
$$F \equiv \frac{d}{dt}\left(\frac{\partial \theta}{\partial y}\right) \qquad (3.1),$$

where, $\theta$ is the temperature. While the frontogenetic function is generally defined as
the square of the horizontal gradient of the temperature (e.g., GC2014), our study
employs only the meridional gradient of the temperature because the ABFZ SST-
gradient is oriented South-North. The right hand side of Eq. 3.1 can be written as,



$$\frac{d}{dt}\left(\frac{\partial\theta}{\partial y}\right) = u\frac{\partial}{\partial x}\left(\frac{\partial\theta}{\partial y}\right) + v\frac{\partial}{\partial y}\left(\frac{\partial\theta}{\partial y}\right) + w\frac{\partial}{\partial z}\left(\frac{\partial\theta}{\partial y}\right) + \frac{\partial}{\partial t}\left(\frac{\partial\theta}{\partial y}\right)$$

$$= -\frac{\partial u}{\partial y}\frac{\partial\theta}{\partial x} - \frac{\partial v}{\partial y}\frac{\partial\theta}{\partial y} - \frac{\partial w}{\partial y}\frac{\partial\theta}{\partial z} + \frac{\partial}{\partial y}\left(\frac{\partial\theta}{\partial t} + u\frac{\partial\theta}{\partial x} + v\frac{\partial\theta}{\partial y} + w\frac{\partial\theta}{\partial z}\right)$$

$$= -\frac{\partial u}{\partial y}\frac{\partial\theta}{\partial x} - \frac{\partial v}{\partial y}\frac{\partial\theta}{\partial y} - \frac{\partial w}{\partial y}\frac{\partial\theta}{\partial z} + \frac{\partial}{\partial y}\left(\frac{d\theta}{dt}\right)$$

$$(3.2)$$

Here, $u$, $v$, and $w$ denote the current velocity and we use the relation between
Lagrangian and Eulerian differentiations. Equation 3.2 describes the processes that act
to generate/destroy the ocean front. The terms $-\frac{\partial u}{\partial y}\frac{\partial\theta}{\partial x}$, $-\frac{\partial v}{\partial y}\frac{\partial\theta}{\partial y}$, and $-\frac{\partial w}{\partial y}\frac{\partial\theta}{\partial z}$ are the
contributions due to the mechanical processes: shear, convergence and tilting,
respectively. The shear term represents conversion of the zonal temperature gradient
into meridional gradient by zonal current shear. The convergence term represents
strengthening/weakening of the meridional temperature gradient by
convergence/divergence of meridional current. The tilting term represents conversion
of the vertical stratification into meridional gradient by meridional shear of vertical
velocity.
The fourth term is a thermodynamical term due to exchange heat associated
with the turbulent heat flux. This term can be expressed as,
$$\frac{\partial}{\partial y}\left(\frac{d\theta}{dt}\right) = \frac{\partial}{\partial y}\left(-\frac{\partial\overline{w'\theta'}}{\partial z}\right) \quad (3.3).$$

The contribution due to the second order horizontal diffusion is ignored for simplicity.
Since within the OML the temperature is fairly uniform (cf. Fig. 2 to compare
the SST and OML-averaged temperature), we consider the OFGF with the mixed-





layer mean quantities. With the approximation that temperature is independent of the
depth in the OML (Kazmin and Rienecker, 1996), Eq. 3.2 can be expressed as,

$$\frac{d}{dt}\left(\frac{\partial \theta_{oml}}{\partial y}\right) = -\frac{\partial u_{oml}}{\partial y}\frac{\partial \theta_{oml}}{\partial x} - \frac{\partial v_{oml}}{\partial y}\frac{\partial \theta_{oml}}{\partial y} - \frac{\partial w_b}{\partial y}\frac{\Delta\theta}{D} + \frac{\partial}{\partial y}\left(\frac{Q_s + Q_b}{\rho C_p D}\right) \quad (3,4),$$

where, the subscript of *oml* indicates the OML-mean quantity. Although the horizontal
velocity is a function of depth even in the OML, the horizontal mechanical terms in
Eq. 3.4 can be written in terms of OML-mean quantities because the production
remains linear relation as long as the temperature is independent of depth in the OML.
$w_b$, $\Delta\theta$ and $D$ represent the vertical velocity, the temperature jump at the bottom of
the OML and the OML depth. We use constant values for sea water density, $\rho$ (1000
kg/m$^3$) and isobaric specific heat of sea water, $C_p$ (4200 Jkg$^{-1}$K$^{-1}$). The vertical mixing
term is replaced with $Q_s$ and $Q_b$, where $Q_s$ is the surface net heat flux at the top of
OML (downward is positive in this study) and $Q_b$ represents the vertical mixing at the
bottom of the OML, i.e., in the thermocline. We assume that there is no penetration of
shortwave radiation beyond the OML to deeper ocean layers. Because the vertical
mixing term expressed by $Q_b$ is a higher-order term, it is expressed as an additional
term; it will be not addressed explicitly in this study.

While Eq. 3.4 is Langrangian form of the OFGF, the equation can be also

expressed in Eulerian form as below:

$$\frac{\partial}{\partial t}\left(\frac{\partial \theta_{oml}}{\partial y}\right) = \underbrace{-\frac{\partial u_{oml}}{\partial y}\frac{\partial \theta_{oml}}{\partial x}}_{SHER} \underbrace{-\frac{\partial v_{oml}}{\partial y}\frac{\partial \theta_{oml}}{\partial y}}_{CONF} \underbrace{-\frac{\partial w_b}{\partial y}\frac{\Delta\theta}{D}}_{TILT} + \underbrace{\frac{\partial}{\partial y}\left(\frac{Q_s}{\rho C_p D}\right)}_{SFLX} + \underbrace{residual}_{RESD} \quad (3.5).$$



The contribution due to the vertical mixing $Q_b$, is estimated as residual of Eq. (3.5).
Along with the vertical mixing, the residual term also includes the horizontal and
vertical advection of the $\partial \theta_{oml} / \partial y$ which are not related to Lagrangian sources of the
frontogenesis either. In the reminder of this paper, the shear term will be referred to as
SHER, the confluence as CONF, the tilting as TILT, the thermodynamic term as
SFLX and the residual as RESD.
Note that basically, our climatology is a 29-years mean from 1982 to 2010.
However, some years do not have OML data at some grid points around the coastal
region. For these grid points, we make the climatology only for available years. For
example, the smallest number in the focusing ABFZ is 16 years at 16.25 °S.

**4. Overview of the ABFZ and its Seasonal Cycle in CFSR data**
Before the dynamical diagnosis is performed, we provide a brief overview of
the main feature of the ABFZ. The maximum of the ABFZ (up to 1.4 °C/100km) is
located at 16 °S just near the coast (Fig.1b). Figure 2a shows a seasonal cycle of the
temperature and its meridional gradient obtained from the satellite product OISST.
The core (SST meridional gradient exceeds 1.0 °C/100km) of the ABFZ always lies
between 17 °S and 15 °S. At seasonal scale, the location of the ABFZ exhibits rather a
weak variability compared to strong interannual variability associated with the
Bengulea Niños that push the ABFZ southward due to the southward intrusion of
tropical warm water (e.g., Gammelsrød et al. 1998; Veitch et al., 2006; Rouault et al.,
2017). For instance, Rouault et al. (2017) shows that during Benguela Niño 2010-
2011 the ABFZ displaced southward as far as 20°S. The intensity of the ABFZ shows
a pronounced seasonal cycle: there are two peaks of the strength in April-May and



November-to-December, respectively. The semi-annual cycle of the ABFZ will be
examined in more details in the following sections. Figures 2b and c evidence that the
CFSR reanalysis reproduces realistically the annual cycle of the ABFZ, and that the
annual cycle of the corresponding OML-mean temperature meridional gradient is
representative of the annual cycle of the SST meridional gradient in terms of both
timing and intensity of the two annual peaks. This latter result justifies our approach
to diagnose the frontogenesis of the ABFZ with the OML-mean quantities.

**5. Dynamical Diagnosis on the ABFZ**

In this section, we investigate the frontogenesis of the ABFZ diagnostically
applying the OFGF described in Section 3. Figure 3 illustrates the climatological
annual-mean oceanic dynamical fields. The southwestward Angola and
northwestward Benguela alongshore currents collide just south of the ABFZ. Seaward
from the ABFZ, a strong westward current is detected. Intense upwelling (vertical
velocity at the bottom of OML exceed 0.18 m/day) is generated along the coast in the
Benguela Current region. A local maximum of upwelling in the ABFZ (approximately
17 °S) corresponds to one of the most vigorous upwelling cells in the region, namely
Cape Frio cell (Lutjeharms and Meeuwis, 1987). Note also a relatively weak
downwelling cell (vertical velocity down to -0.06 m/day) just seaward from the Cape
Frio upwelling cell.

*5.1 Annual-mean state*





Figure 4 presents the annual-mean climatology of the 5 forcing/source terms
of the OFGF superimposing the meridional gradient of the OML-mean temperature.
SHER works frontolytically (destroying the front, about -2 °C/100 km×$10^{-7}$ s$^{-1}$) in the
most parts of the ABFZ except just near the coast at 17 °S, although its frontogenetic
(generating front) contribution here is rather weak (less than 2 °C/100 km×$10^{-7}$ s$^{-1}$).
CONF has on average an intense frontogenetic contribution to the ABFZ (up to
5 °C/100 km×$10^{-7}$ s$^{-1}$), especially offshore around 16 °S where the ABFZ is centered
(Fig. 2). The frontogenetic effect of CONF is consistent with GC2014 (the
frontogenesis of the SST front associated with the equatorial Atlantic cold tongue is
due to the confluence of northern South Equatorial Current and Guinea Current) and
can be expected because the warm and cold currents meet around the ABFZ. Note
however a small zone just near the coast at 16 °S where the CONF is frontolytic. This
local frontolytic contribution is overcompensated by a strong frontogesis due to TILT
(more than 5 °C/100 km×$10^{-7}$ s$^{-1}$ on average in the ABFZ core). An elongated
frontogentic zone associated with TILT is found along the Angolan coast from 17°S to
11°S and corresponds to the upwelling tongue observed in the Angola current region
(Fig.3). On the other hand, TILT is frontolytic off the ABFZ (at 17°S, 11°E) where the
downwelling is dominant as shown in Fig.3. The role of the upwelling in the ABFZ
development will be analyzed in more details in the Section 6.2.
In addition to the mechanical terms, the thermodynamical components also
show some influences on the ABFZ. SFLX works frontogenetically just near the coast
at 16°S and frontolytically south and north from the core of the ABFZ, although its
contribution is almost negligible compared to the mechanical contribution. RESD is
estimated by,



$$\text{RESD} = \frac{\partial u_{oml}}{\partial y}\frac{\partial \theta_{oml}}{\partial x} + \frac{\partial v_{oml}}{\partial y}\frac{\partial \theta_{oml}}{\partial y} + \frac{\partial w_b}{\partial y}\frac{\Delta\theta}{D} - \frac{\partial}{\partial y}\left(\frac{Q_s}{\rho C_p D}\right) \quad (5.1)$$

where, we assume that there is no local temporal tendency of the front, $(\partial\theta_{oml}/\partial y)/\partial t$
so that Eq. (3.5) can be closed. One annual time scale this approximation is robust. On
average in the core of the ABFZ, RESD shows a strong frontolytic contribution
around the core of the ABFZ (Fig. 4e). On the other hand, frontogenesis is located in
the southern part of the ABFZ. This may be due to, at least, to vertical mixing at the
base of the OML accounted for in RESD. According to GC2014, the turbulent mixing
(surface and thermocline heat fluxes) is frontolytic in the equatorial front.

*5.2 Seasonal Cycle*
In the preceding subsection, we have shown that in terms of climatological
annual-mean terms CONF and TILT of the OFGF are the main drivers for the ABFZ
generation. Next, we analyze the annual cycle of the ABFZ and its relationship to the
seasonal variations of the OFGF terms. Note that Eq. 3.5 implies $\pi/2$ out of phase
between the OFGF and temperature meridional gradient. This means that for a semi-
annual oscillation the temperature meridional gradient should lead the OFGF by
approximately 1 and half months.
Figure 5a illustrates the box-mean (10 °E-12 °E and 17 °S-15 °S) temporal
series of the meridional gradient of temperature obtained from satellite and reanalysis
products (the time series is smoothed by a 11-days-mean moving filter). There is an
obvious semi-annual cycle of the ABFZ with maxima in April-May and in November-
December, respectively, and minima in February-March and July-August,



respectively (see also Fig.2). The first maximum develops rapidly (during 2 month,
from March to April) whereas the development of the second maximum is somewhat
slower (3 months, from August to October). Figure 5a also evidences that CFSR
reproduces realistically the semi-annual cycle, although the magnitudes of the CFSR
meridional SST gradient are generally slightly stronger with respect to OISST.

We further analyze the seasonal cycle of the OFGF terms. Similarly to the

climatological state in Fig. 4, the contributions of SHER and SFLX are relatively
small and do not seem to be responsible for either of the two peaks in the ABFZ
annual cycle (not shown). Figure 5b shows the seasonal variations of TILT, CONF,
and RESD averaged over the same box as the temperature gradients in Fig. 5a. For
estimation of seasonal variation of RESD, the tendency of the meridional gradient is
calculated as,
$$\frac{\partial}{\partial t}\left(\frac{\partial \theta_{oml}(t)}{\partial y}\right) = \frac{\dfrac{\partial \theta_{oml}(t+1)}{\partial y} - \dfrac{\partial \theta_{oml}(t-1)}{\partial y}}{2 \times \text{Day}}, \quad (5.2)$$

where, $t$ denotes each time step, in this case, daily. With this tendency at each day,
RESD($t$) is estimated by
$$\text{RESD}(t) = \frac{\partial}{\partial t}\left(\frac{\partial \theta_{oml}(t)}{\partial y}\right) - \text{SHER}(t) - \text{CONF}(t) - \text{TILT}(t) - \text{SFLX}(t).$$

From the middle of November to February, the box-averaged CONF is

modestly negative, which is due to the frontolytic effect adjacent to the Angolan coast
as shown in Fig. 4b (however, CONF is frontogenetic off the ABFZ). The
contribution of CONF becomes positive from March, although its frontogenetic
contribution is relatively weak ($< 1.0$ °C/100 km$\times 10^{-7}$ s$^{-1}$) until July. From the end of



July CONF starts to increase and reaches its maximum ($3.0$ °C/$100$ km$\times10^{-7}$ s$^{-1}$) in the
end of August. The frontogenetic contribution of CONF remains strong until the
beginning of October but then rapidly decrease to become frontolytic in November.

The contribution of TILT to the ABFZ seasonal cycle is almost always

frontogenetic. Close to zero in January, TILT is enhanced from February and reaches
its maximum value ($3.0$ °C/$100$ km$\times10^{-7}$ s$^{-1}$) in March-April. In May-June, the
frontogenetic effect of TILT gradually decreases (down to $1.0$ °C/$100$ km$\times10^{-7}$ s$^{-1}$) until
December. The maxima in TILT and CONF correspond to the two periods of
development of the ABFZ at seasonal scale: from March to April and from August to
October, respectively (Fig. 5a). This suggests that the two peaks of the ABFZ are
associated with two different mechanical terms and thus are due to two different
physical processes. On the other hand, the two periods of decay of the ABFZ are
consistent with the periods of weak frontogenetic and/or frontolytic contributions of
both TILT and CONF, in December-February and June-July, respectively.

In addition, RESD is almost always frontolytic with a relatively large

oscillation ($0.0$ to $-5.0$ °C/$100$ km$\times10^{-7}$ s$^{-1}$) as shown in Fig.5b. In particular, the
frontolytic effect due to RESD is stably strong (around $-3.0$ °C/$100$ km$\times10^{-7}$ s$^{-1}$) from
May to August when the ABFZ becomes weakened and frontogenetic effects due to
CONF and TILT are relatively weak (Figs. 5a and b). Conversely as TILT and CONF,
RESD does not exhibit a clear signal of semi-annual cycle, but rather an annual-cycle.
We thus can conclude that in terms of a first-order estimation, the semi-annual cycle
of the ABFZ is explained by the combination of TILT and CONF.

**6. Discussion**



The previous section showed that the two periods of development of the
ABFZ in March-April and August-October were due to a large extent to the
contribution of TILT and CONF, respectively. In this section, we investigate what
components are responsible for the corresponding peaks in TILT and CONF.

*6.1 Meridional Confluence*
CONF represents changes in the meridional temperature gradient associated
with ocean dynamics of convergence/divergence of meridional current, $\partial v_{oml} / \partial y$.
Figure 6a presents the annual cycle of $\partial v_{oml} / \partial y$ averaged over the ABFZ that shows a
mirror image of the time series of CONF (Fig. 5). In the ABFZ, the meridional current
is almost always convergent except for weak divergence from November to January.
The convergence of the meridional current is maximum from August to mid-October
(up to $-3.0 \times 10^{-7}$ s$^{-1}$) and is rapidly weakened during November. The seasonal
fluctuations in the convergence are associated with changes in intensity and
meridional extension of the southward Angola Current and northward Benguela
Current that meet in the ABFZ. Figure 6b illustrates the annual cycle of OML-mean
meridional current and meridional component of geostrophic current estimated from
sea surface height (SSH) at 15 °S (north of the core of the ABFZ) and 17 °S (south of
the core of the ABFZ) averaged between 10 °E and 12 °E. At 15 °S the OML-mean
meridional current is southward all year round, except the beginning of May when a
weak northward flow is observed. The maximum southward meridional velocity
occurs in October (-0.12m/s). At 17 °S the OML-mean meridional current is
northward in March-June and shows a bi-annual peak of southward current in
January-to-mid-February and October indicating intrusion of tropical warm water to



the ABFZ (e.g., Rouault, 2012). Figure 6b clearly evidences that the region between
17 °S and 15 °S is expected to be convergent. The most convergent period is in
September-October when the CONF contribution to frontogenesis is the largest as
shown in Fig. 5b. Another relatively strong convergent period is from April to June
when the meridional current is rather northward at 17 °S and close to zero at 15° S.
The period of weak convergence/divergence, form December to February,
corresponds to frontolytic contribution of CONF (Figs.5b). Figure 6b evidences that
the OML-mean meridional current can be explained, to a large extent, by the
geostrophic surface current.
The spatial distributions of the climatological monthly mean SSH and surface
geostrophic current in January, April, and September are shown in Figure 7. Two local
minima of SSH are observed: one along the coast in the Benguela system and one
west of the ABFZ (centered at 14 °S and 6 °E). The latter is associated with the
Angola Dome (e.g., Doi et al. 2007) and a strong cyclonic geostrophic flow reaching
the ABFZ. The geostrophic current generally generates the convergence in the ABFZ
(Fig. 6a). However, in January an intense divergence is generated due to the strong
southward ageostrophic current along the coast (Fig. 7a). In April, when CONF is
modestly frontogenetic (Fig.5b), the Angola Dome and associated geostrophic flow
are diminished (Fig. 7b) and a main source of convergence can thus be attributed to
the northward Benguela Current which penetrates into the ABFZ as far as up to 16 °S.
In September, whereas the low SSH sits in the south of the ABFZ as in April, the
Angola Dome is significantly developed to induce a strong geostrophic current
resulting in a strong southward Angola Current intruding into the ABFZ along the
Angolan coast. The northward Benguela Current is relatively weak in September



compared to that in April. Thus, the maximum CONF in September is due to the
strong southward Angola Current.

*6.2 Tilting*

TILT is the second main contributor to generate the ABFZ especially in

March-to-May as shown in Figs. 4 and 5. In a first approximation TILT results from
the meridional gradient of vertical motion $\partial w_b / \partial y$ convoluted with the thermocline
stratification (e.g., Eq.3.5). Here, we explore more details of upwelling in the ABFZ.
The annual cycle of these two components averaged over the box [12 °E-10 °E] and
[17 °S-15 °S] (Fig.8) points out that $\partial w_b / \partial y$ and the stratification is negative and
positive, respectively, from January to August. This configuration leads to
frontogenesis through the TILT term (Fig. 5b). From August to December, $\partial w_b / \partial y$
changes sign and the stratification becomes weaker; that explains why the TILT term
is frontolytic (especially in September) and its magnitude is weaker compared to
January-August because of a weaker stratification. Negative $\partial w_b / \partial y$ can be seen in
both March to April and August to September around the ABFZ in Figs. S1a and b,
but positive $\partial w_b / \partial y$ are also generated around the ABFZ more in August-September
than in March-April.

The OML depth has extrema in August to September (around 100 m) and from

January to April (around 20 m) indicating the seasonal cycle of solar insolation
forcing. Also the intensity of the thermocline shows a strong stratification from March
to May (2°C) and weak stratification from September to November (1.2°C). From
March to May TILT is the most dominant frontogenetic source because the OML is





the shallowest (20-30m), the stratification is the strongest (up to 2.0K) and the shear
of vertical velocity $\partial w_b / \partial y$ is strongly negative. The shallow OML and strong
stratification can amplify the tilting effect due to $\partial w_b / \partial y$. Conversely, TILT is weakly
frontolytic from August to September when the OML-depth is deepened (~100m), the
stratification is weak (1.2K) and $\partial w_b / \partial y$ is positive. Fig.S1c and d shows the
differences in OML depth and ocean stratification between March-April and August-
September. Shallower OML and stronger stratification can be seen everywhere around
the ABFZ. Therefore, ~~the~~ effects of both positive and negative $\partial w_b / \partial y$ are reduced
and consequently, contribution of TILT is quite weak in August to September (Fig.
5b).

**7.  Concluding Remarks**

In this study we investigated the processes controlling the ABFZ evolution

based on a first-order estimation of an ocean frontogenetic function (OFGF) applied
to the ocean mixing layer (OML) derived from the CFSR reanalysis. The OFGF
represents the temporal evolution of the meridional mixed-layer temperature gradient
and contains three mechanical terms (shear, convergence and tilting) and one
thermodynamical term. The residual term accounts for higher-order terms, associated
in particular with vertical mixing and horizontal/vertical advections of the meridional
temperature gradient. An analysis of the annual mean OFGF suggests that the
confluence effect (CONF) due to southward Angola Current (warm) and northward
Benguela Current (cold) is dominantly frontogenetic over the offshore part of the
ABFZ, although it has a local frontolytic effect just near the coast at 16ºS. The tilting
effect (TILT) related to the coastal upwelling regime is another main contributor to





frontogenesis..The contributions of the shear (SHER) and surface heat flux (SFLX)
terms, are rather negligible, while the residual (RESD) represents a main frontolytic
source.

Seasonal evolution of the ABFZ has a well-pronounced semi-annual cycle

with two maxima, in April-May and November-December, and two minima, in
February-March and July-August. We showed that the two maxima of the ABFZ were
associated with two different mechanical terms and due to two different physical
processes. The development of the first ABFZ maximum during March-April is
mainly explained by the strong contribution of TILT to frontogenesis, while the
development of the second ABFZ maximum during September-October is due to the
frontogenetic contribution of CONF. TILT is associated with the merdional gradient
of the vertical velocity.  The annual maximum of TILT in March-April is due to a
large extent to the combination of the maximum stratification ($\Delta\theta$), shallow OML
depth ($D$) and negative $\partial w_b / \partial y$ during this period. Indeed, in OFGF formulation the
ratio $\dfrac{\Delta\theta}{D}$ represents the efficiency by which the meridional gradient of the coastal
upwelling velocity can lead to the change of the ABFZ intensity. Although the OML
depth also modulates the surface heat flux contribution to the OFGF, the
thermodynamical term does not show any significant impact on the development of
the ABFZ maximum in March-April. On the other hand, the importance of the OML
depth for the thermodynamical term was suggested for frontogenesis in a SST front
associated with western boundary current (Tozuka and Cronin, 2014; Tozuka et al.,
2018). The annual maximum of CONF in September-October is related to an
intensified southward Angola current that seems to be induced by a cyclonic
geostrophic flow associated with the development of the Angola Dome (e.g., Doi et



al., 2007). A relatively smaller contribution of CONF to frontogenesis is also observed
in April and is due to the intrusion of the northward Benguela Current to the ABFZ
during this period.

Most CGCMs fail to reproduce realistic SST field and ABFZ location. Among

other causes, this can be due to a poor representation of regional climate variables in
CGCMs, such as upwelling favorable wind, near-coastal wind curl and wind drop off,
alongshore stratification and OML depth (e.g., Xu et al., 2014; Koseki et al., 2018;
Goubanova et al., 2018), that impact directly the two main frontogenesis terms,
CONF and TILT.  The OFGF proposed in the present study can be thus an appropriate
tool to diagnose the performance of CGCMs in the ABFZ and more generally in
frontal zones. This study shows that diagnosis developed for mesoscale studies are
valuable for climate studies and can help to identify the origin of biases which affect
OGCMs. Effects of the turbulent mixing at the mixed-layer base on frontogenesis
were accounted by the residual of the frontogenetic function. This is the main
limitation of this study because diapycnal mixing is often an important term of the
oceanic upper-layers heat budget which is tightly coupled with vertical motions
(Giordani et al., 2013). A more comprehensive understanding of this term would be
valuable to estimate the performance of CGCMs in the ABFZ and more generally in
coastal upwelling zones.


**Acknowledgement**
We greatly appreciate Dr. Kunihiro Aoki in the University of Tokyo for his
constructive discussion in the beginning of stage of this study. We also thank to Dr.



Guy Caniaux in MÉTÉO-France for their helpful discussions. We utilized the versions
of 2012Rb of MATLAB software package provided by The MathWorks, Inc.,
(http://www.mathworks.com) and Grid Analysis and Display System (GrADS,
http://www.iges.org/grads/) to compute each dataset and create figures. The research
leading to these results received funding from the EU FP7/2007-2013 under grant
agreement to no. 603521 (EU-PREFACE).

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

**Figures**

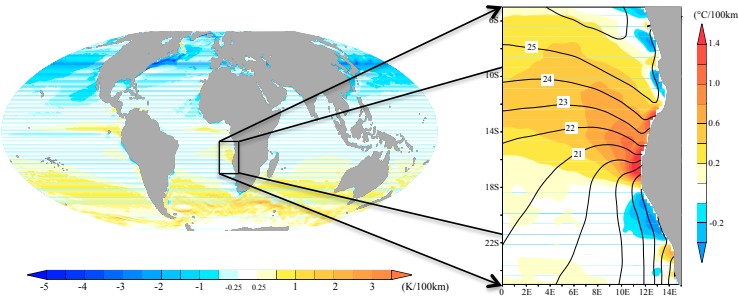

**Figure 1**.
(Left) Global image of observed annual-mean SST meridional gradient from 1982-2010 of OISST. (Right) annual-mean SST (contour, °C) and its meridional gradient (°C/100km) around the ABFZ.






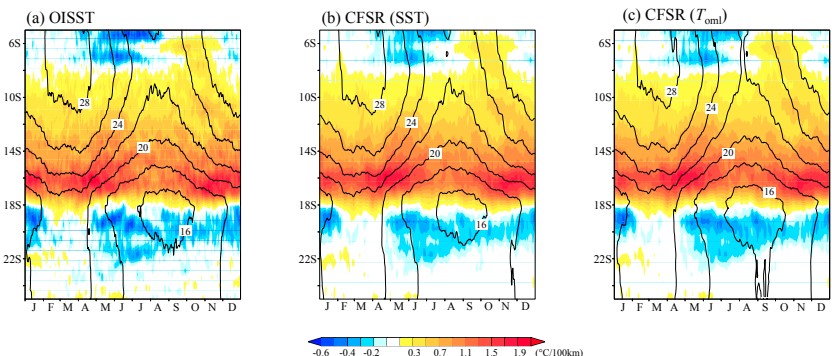

**Figure 2.**
Climatological seasonal cycle of the temperature (contour) and its meridional gradient averaged
between 10°E and 12°E for (a) SST of OISST, (b) SST of CFSR, and (c) OML-mean potential temperature of CFSR.


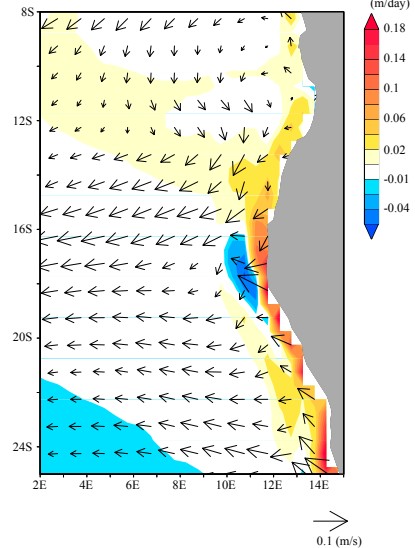

**Figure 3.**
Annual-mean climatological states of OML-mean horizontal current (arrows) and vertical velocity
at the bottom of OML (color).






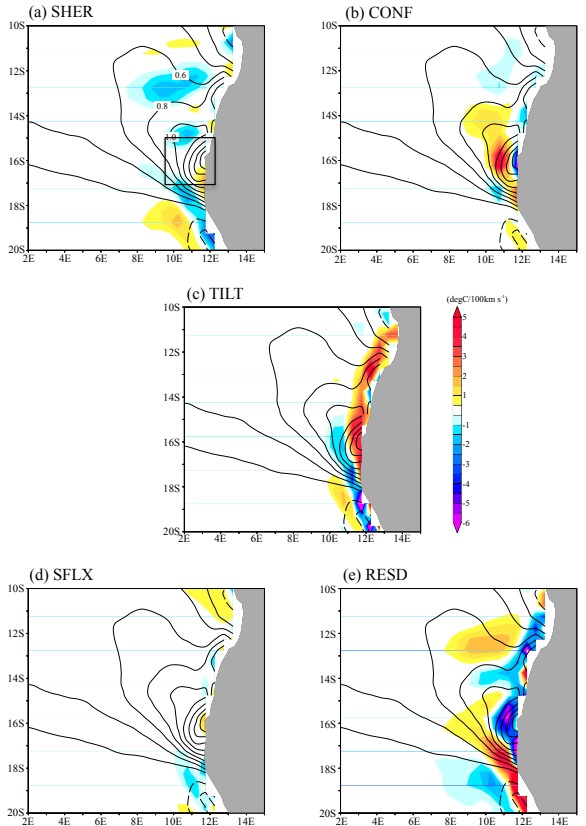

**Figure 4**.
Annual-mean climatology of each term in OFGF. Contour is annual-mean climatology of meridional gradient of OML-mean potential temperature of CFSR (°C/100km). The black box on (a) is the ABFZ used for the analysis in this study.







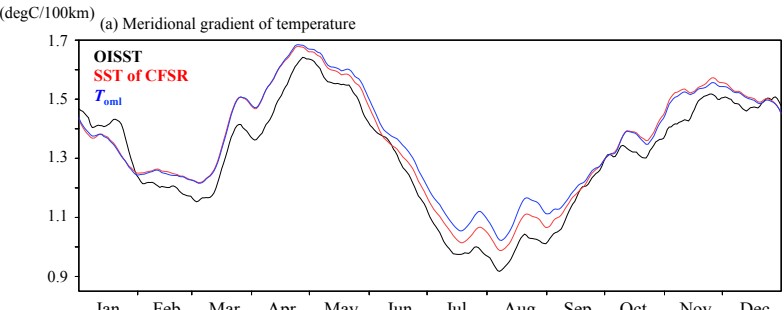

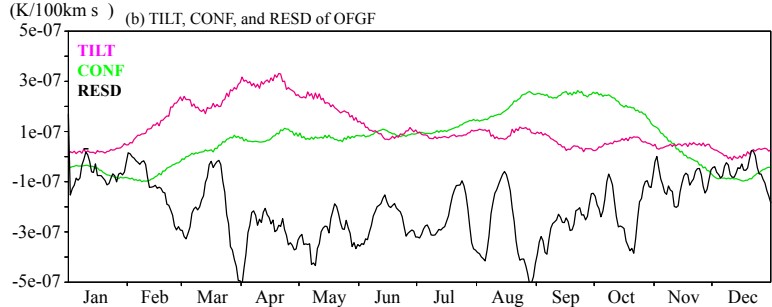

**Figure 5**.
Box-mean (17°S-15°S and 10°E-12°E) time series of (a) meridional gradient of temperature (black: OISST, red: SST of CFSR, and blue: OML-temperature of CFSR) and (b) TILT (magenta), CONF (green) and RESD (black). 11days-running mean are shown for all the time series.





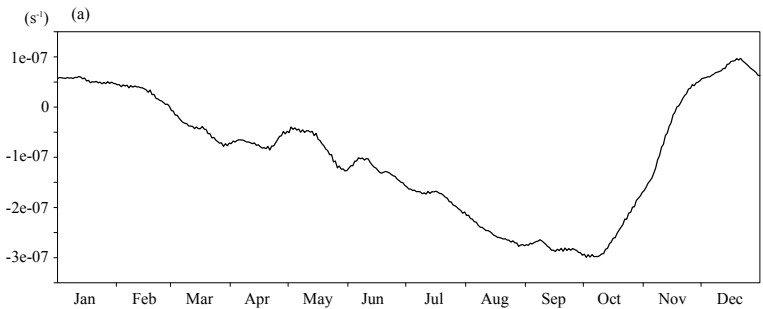

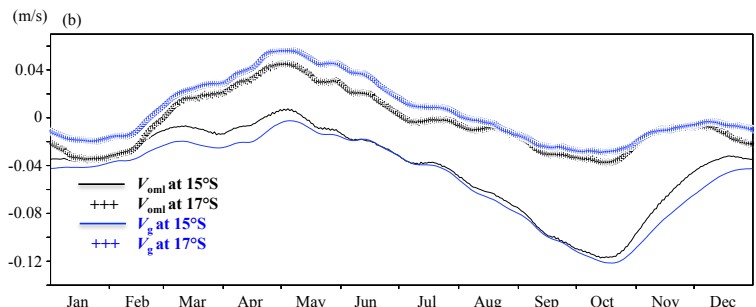

**Figure 6**.
Time series of (a) $\partial v_{oml} / \partial y$ averaged over (17°S-15°S and 10°E-12°E ) and (b) OML-mean meridional current velocity (black) and geostrophic meridional current velocity estimated from sea surface height (blue) at 15°S (solid line) and 17°S (+ mark) averaged between 10°E and 12°E. All variables are filtered by moving 11-days window.




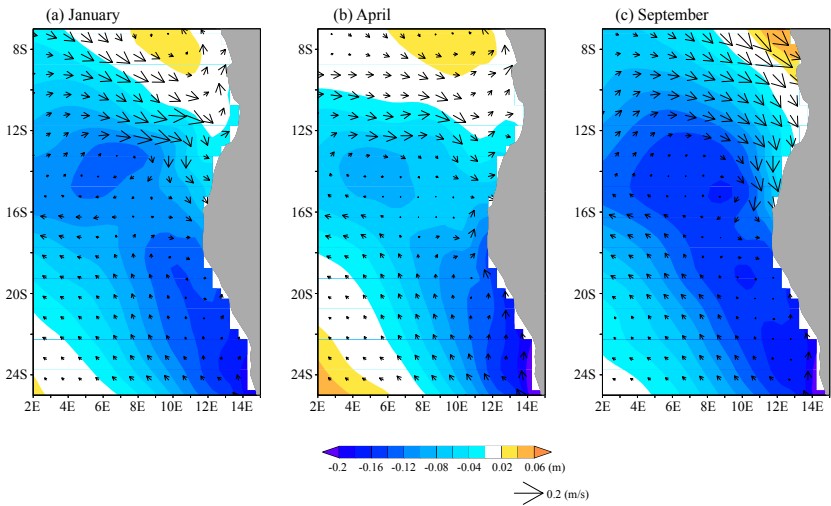

**Figure 7**.
Monthly mean SSH (color) and geostrophic current (arrows) for (a) January, (b) April, and (c)
September.


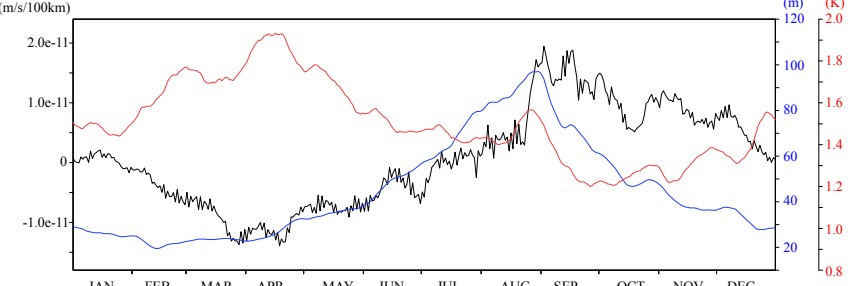

**Figure 8**. Time series of the area-averaged meridional gradient of the vertical velocity at the bottom of
OML (black), OML depth (blue), intensity of upper ocean thermocline stratification (red) over
17°S-15°S and 10°E-12°E. All variables are filtered by moving 11-days window.

