# Peer review of "Frontogenesis of the Angola-Benguela Frontal Zone"

_Ocean Science, 2018_

## Referee Comment (RC1) · Anonymous Referee #1 · 1 Oct 2018

General comments:

I have been following studies in this region and I was always wondering about the generation mechanism of the Angola-Benguela Frontal Zone (ABFZ). As such, I read this manuscript with great interest. In this paper, the authors showed that the strength of ABFZ undergoes semiannual variation with the maxima in Apr.-May and Nov.-Dec., but the mechanism for the two peaks is quite different. The first maximum is due the tilting term, while the second maximum is due to the confluence term.

Although this manuscript presents new insight into the seasonal variation of the ABFZ, there are some issues with their frontogenesis equation. Therefore, I recommend publication of this manuscript after major revision.

[Figure]

Major comments:

1) Equation (3.4): To obtain Eq. (3.4) from Eq. (3.2), we need to integrate Eq. (3.2) from the surface to the bottom of the OML and divide by the OML depth. If the OML depth were not constant in time and space, terms containing the derivatives of the OML depth should appear, but those terms are missing in Eq. (3.4). Please check the appendix of Moissan and Niiler (1998), for example.

Moisan, J. R., and P. P. Niiler (1998), The seasonal heat budget of the North Pacific: Net heat flux and heat storage rates (1950-1990), J. Phys. Oceanogr., 28, 401-421.

2) Equation (3.5): The assumption that there is "no penetration of shortwave radiation beyond the OML to deeper ocean layers" may not be a good assumption considering that the OML depth could be as shallow as 20 m in the region during austral summer.

3) How is the OML depth determined? Is it based on some density criterion? Also, how is delta theta calculated.

4) Lines 389-392: Since the authors are using a reanalysis product, effects of data assimilation are also included in the residual term.

5) Figure 5b: The tendency term should also be shown.

Minor comments:

1) Figure 1: Label the left panel as Fig. 1a and the right panel as Fig. 1b (Line 169).

2) Although the editor provided many comments on their writing, there are still some grammatical errors etc.

Line 10: Replace "form" with "from".

Line 11: Delete ", respectively".

Line 30: Replace "Resason" with "Reason".

Line 36: Delete "over the southern African Continent" as this phrase is redundant.

Line 38: Replace "model" with "mode".

Line 49: Replace "GCM" with "CGCMs".

Line 57: Replace "insightfully" with "quantitatively".

Line 62: Replace "at" with "in".

Line 70: Replace "opposite " "opposing".

Line 72: Replace "more" with"most".

Line 86: Replace "put" with "make".

Lines 92-93: Replace "National Ocean and Atmosphere Association" with "National Oceanic and Atmospheric Administration".

Line 119: Replace "current velocity" with "zonal, meridional, and vertical current velocity, respectively,"

Line 129: Add "of" after "exchange".

Line 138: Replace "3,4" with "3.4".

Line 143: Replace "velocity," with "velocity and".

Line 144: Add ", respectively" after "depth".

Line 158: Replace "reminder" with "remainder".

Line 161: Replace "29-years" with "29-year".

Line 172: Delete "a" before "weak".

Line 176: Replace "shows" with "showed".

Line 193: Replace "exceed" with "exceeding".

Line 227: Replace "One" with "On".

[Figure]

Line 238: Add "relationship" after "phase".

Line 242: Replace "temporal" with "time".

Line 244: Replace "days" with "day".

Lines 246 and 247: Delete "respectively".

Line 247: Replace "month" with "months".

Line 311: Add "in" after "except".

Line 321: Replace "form" with "from".

Line 322: Replace "Figs." with "Fig.".

Line 349: Replace "is" with "are".

Line 356: Replace "are" with "is".

Line 367: Replace "Fig." with "Figures" and "shows" with "show".

Line 387: Delete one period.

Line 406: Replace "a" with "an".

Line 409: Replace "current" with "Current".

Line 421: Replace "are" with "is".
* * *

---

## Referee Comment (RC2) · Anonymous Referee #2 · 19 Oct 2018

As said in the papers introduction, a quantitative analysis of the frontogenesis in the ABFZ is really not done yet. So, the paper deserves interest and may enhance our knowledge on the dynamics of the boundary between tropical and subtropical waters in Eastern Boundary Currents of the southern Atlantic. So I recommend to publish it. However, I propose to revise the paper first, for two reasons. It has capacity to gain better scientific quality at some points and some steps in the reasoning are either not well described or not justified. The main crititics are on the way how the "residual" term is treated.

Let us go through the paper in detail. Remarks are made in the "order of appearance". This mixes important and less important ones.

40) Does the SST and the ABFZ really influence the rainfall activity or are anomalies

of both related to the same process?

52) To reduce the SST bias in model results, a more realistic wind forcing is needed. This would reduce the overestimated poleward transport of tropical water. Understanding the ABFZ dynamics only does not reduce the model bias. But for sure, understanding something helps to deal better with it. It was a major outcome of the PREFACE projects that the SST bias is mostly due to shortcomings on the atmosphere side.

56) I think, Mohrholz et al. 1999, already gave some insight in the frontal "dynamics" in that sense that processes moving the front are identified. But you are right, a quantitative discussion in terms of hydrodynamic equations was not given there.

71) Just to identify the "current systems", "Benguela current" and "Angola current", is not a discussion of the dynamics. The Benguela current has a more complex structure, one part that should contribute to the frontal dynamics, is the wind driven coastal jet. For the Angola current coastally trapped waves are important.

73) What is meant here? How does the specific cloud cover modify the heat flux components? Is it reducing or enhancing influence?

99) .5 deg resolution is not really good? There are other ocean model products available with better horizontal resolution, e.g. from the ECCO group. What is the motivation of your choice?

102) This justification for using the potential temperature does not hit the point. Please look into some textbook (say Olbers et al., Ocean Dynamics) for the in-situ temperature transport equations. It is a highly complex one. The corresponding analysis of the frontogenetic terms would be a nightmare. Potential temperature or conservative temperature do not stand for a the thermodynamic quantity "temperature". But you are right, potential temperature at the surface is numerically equal to the in-situ temperature. Hence, potential temperature is suitable for your analysis and results can be presented as "temperature". Moreover, the approximate transport equation looks like

a classical advection diffusion equation. May be you just say that you are considering potential temperature? Thats it.

119) What is the relation between Lagrangian and Eularian differentiations? As far as I see just commutation and product rule of differential calculus are considered.

121) In the beginning, you made the very basic assumption that the front orients zonally. Now you consider the transformation of zonal gradients into meridional gradients. Do you consider zonal temperature gradients or do you neglect them? You basic assumption needs a better justification or zonal components of the fronts should be included? My view is that a major contribution to frontogenesis is upwelling and equatorward transport of cold water with the coastal jet. So zonal temperature gradients are very important. See the images in Mohrholz et al. 1999!

131) The equation is not correct. The vertical divergence of the solar radiation is missing.

141) What do you mean with "production remains linear relation". I guess you mean that as long as the temperature is constant within the OML, the vertical averaging over the horizontal velocity can be carried out. Vertical mixing is not "replaced" by the heat fluxes, it is the upper boundary condition of the integrated mixing term. $Q\_b$ is the lower boundary condition for the vertical heat flux.

150) Please explain in which sense the mixing at the bottom is "higher order". In Fig. 4 the amplitude of the residual term is the largest of all contributions.

161) This needs to be clarified. Which climatology? Do you calculate velocity and temperature climatologies first and from this the different contributions? Gaps in the model results? How can this be?

170) I would propose to make clear somewhere that "strength of the ABFZ" or "intensity of the ABFZ" means the maximum value of the meridional SST-gradient, doesn't it?

187) Here it must be said that "dynamics" means analysis in terms of forces and responses. This is not done is this paper. Considering time changes or trajectories without considering the reasons, i.e., the forces, I would speak about "kinematics". In oceanography is becomes colloquial to say "dynamics" to address "time development". This is not correct in the original meaning of the word.

196) Today this cell is mostly called "Kunene upwelling cell", which is related to a very persistent wind patch off the mouth of river Kunene. Please check this.

225) This is not RESD of eq. (3.5). Please explain better.

226) If you calculate the time average, the result is independent of time. Your reasoning is not convincing. Please describe, what you are really doing and how the quantities displayed in the figures are really calculated. Why don't you just calculate the residual term in 3.5? If you integrate the left hand side of 3.5 in time, you get the difference of the meridional temperature gradient between end and beginning of your time series. Is this difference really zero or small? You said earlier that the interannual variability is much higher than the seasonal variability. So, please add a figure that this difference is smaller than all the other terms. If this is not the case, your RESD is wrong.

236) As said before, TILT and CONF are kinematic terms. So they do not "drive" the generation of the front. Something drives the currents causing tilt and confluence. This is mostly the wind (local and remote) and vertical heat fluxes. But for sure TILT and CONF are the main contributions to the OFGF.

238) The phase shift is an interesting point and needs more attention.

242) The box integral stands for the time derivative of the north-south temperature difference. This could be, one may assume this, a constant, but the temperature gradient inside the box may vary between a linear one to a sharp front inside the box. Is the average over the box a good measure for the overall strength of the ABFZ? The maximum gradient, a median of the gradient or the meridional temperature variance may be also an interesting quantity.

259) Please do not mix variables and units. I propose to introduce \Delta t and say this quantity is 1d.

281) The "two different physical processes" are a very important hypothesis. I think that the paper of Mohrholz et al. (1999) shows the action of TILT and CONF in a descriptive manner. In the new paper we see the equations.

300) I cannot see the "mirror image".

302) To see this, a graticule or some "zero line" should be added in Fig. 5 and 6.

309) Which SSH? From the model? How did you calculate OML-depth? Why only the geostrophic component? Some more details in the "data"-section would be of great help. More important, the concept of reasoning should be made clear. So, what is the motivation, to consider geostrophy only? Fig.7 does not show the Ekman transport, which should be an essential part of the surface flow in the trade wind zone. I would expect it as divergent from the pronounced wind stress curl. If not, the model would be fully Sverdrup balanced away from the coast? Please add some remarks on your way of reasoning and comment.

336) A geostrophic balance between a pressure gradient and the Coriolis force of a current does not imply any information on the driving mechanisms. The Angola Gyre may be related to the wind stress curl in that area but the doming structure and the currents develop together. For me it seems to be more reasonable that the dome is result of a flow, but for sure not of the geostrophically balanced one. It is not intension of this paper to explain the dynamics of the dome. Just, please avoid oversimplified unphysical wording. Proposal: replace "to induce" by "related to".

349) what is negative stratification? In Eqs. 3.4 and 3.5 the sign convention of \Delta \theta is not explained. I tried to understand 349/350. Does it mean negative dw/dy and negative stratification (whatever this is) from January to August and positive dw/dy and positive stratification otherwise? Also 354 to 357 leaves me confused. Aug-Sep.

dw/dy is negative but is positive from August to December. Not quite clear. What is Fig. S1a/b? It cannot be Fig. 1 a/b since this does not show w.

360) What about wind driven mixing?

367) Where are the Figures?

372) This was a chapter on vertical velocity. So, I would have upwelling in my mind, but the chapter closes without even mentioning "wind". So, as a discussion, it remains very technical and less physical. The discussion is mostly on the kinematics of the front and not on the dynamics (in terms of forces).

380) higher order with respect to which parameter?

381) Do you mean "vertical mixing at the bottom of the OML"?

391/2) two maxima of what? Maximum gradient?

414) The variability of the frontal position is not discussed in this paper. However, the common view on the ABFZ variability is that of a seasonally varying meridional movement of the fronts. The stripe between 17°S and 15°S is relatively narrow. Please show that you cover the major part of the ABFZ during all model years.

Fig. 2) Averaging from 10°E to 12°E excludes the coastal areas north and south of 17°S. Fig. 4 shows that this may be a poor approximation. The same applies for other figures using the same approximation. I propose to select a stripe along the coast, probably with a width of some Rossby radii?

Fig. 4) I propose to add a figure showing the time dependent term. It should become small for a long averaging period. But this should be shown. Somewhere in the text the strong interseasonal variability was mentioned. So it could be that this term is large. If the interannual variability is larger than the seasonal cycle, how significant is this figure? For me it is not really satisfying to see that RESD is the largest contribution. I would suggest to write down the details on how the averaged terms (as well as the

monthly climatologies) are calculated from the daily model output. I think the only term, which cannot be calculated is that related to vertical diffusion at the base of the OML. Even here at least the order of magnitude could be estimated.

Fig 5b) I am missing the other contributions. A zero line would be of some help. The same applies for the following figures.

In summary the paper is of great interest. The message is that just "normal" linear advection and mixing even with coarse resolution may generate a frontal system. However, the word "dynamics" is to promising. The paper considers kinematiks of the fronts, the consideration of dynamics, i.e., a discussion of the relation of the development of the front in relation to atmospheric or other driving forces is still outstanding. Unfortunately the typical wind pattern are neither mentioned nor included in the discussion. However, we should not expect that the authors do in one paper what a whole community could not accomplish within decades. So my proposal would be, not to include an analysis of the winds here and just to consider the paper as a step into this direction.

More details on the data processing are needed, especially on the time averaging to gain the climatology. Sometimes I suspected that contributions to the OFGF are calculated from climatological quantities. Please make a clear statement on the method.

The residual term is to large to be called residual. The remaining terms collected here should get a name and deserve more detailed discussion. What about a \theta/D dw/dx term? The reasoning that the left hand side of Eq. 3.5 is small, which defines RESD (Eq. 5.1) is not convincing. This problem should be reconsidered, Eq. 5.1, if it holds, requires more justification.

The fact that the confluence of the geostrophic flow is the major contribution to the CONF-term is important and should be discussed more explicitly. Fig. 7C shows a strong divergence of the geostrophic flow. The water must stay somewhere. Here it continues most probably as the zonal component of the Ekman transport. Since this term is not in the OFGF, it does not become visible here. So one may conclude that a

consideration of geostrophic currents may be sufficient to understand the time evolution of the frontal system. I am not convinced that this is true. Please consider this issue.

---

## Author Comment (AC1) · 5 Dec 2018

Reply to Reviewer#1

**General comments:**
**I have been following studies in this region and I was always wondering about the generation mechanism of the Angola-Benguela Frontal Zone (ABFZ). As such, I read this manuscript with great interest. In this paper, the authors showed that the strength of ABFZ undergoes semiannual variation with the maxima in Apr.-May and Nov.-Dec.,but the mechanism for the two peaks is quite different. The first maximum is due the tilting term, while the second maximum is due to the confluence term. Although this manuscript presents new insight into the seasonal variation of the ABFZ, there are some issues with their frontogenesis equation. Therefore, I recommend publication of this manuscript after major revision.**

We greatly appreciate the reviewer for his/her quite constructive and helpful comments on our manuscript. As below, we have replied to all the comments and added more figure, sentences, and descriptions in the revised manuscript. Please note that the corrections in the revised manuscript are shown by **blue-colored font**.
Please note that we changed the title of this manuscript to "**Frontogenesis of the Angola-Benguela Frontal Zone**" replying to the reviewer's comments of Reviewer#2. Also, please note that the number of equation in section 3 decreases from 5 to 4 following revisions.

**1) Equation (3.4): To obtain Eq. (3.4) from Eq. (3.2), we need to integrate Eq. (3.2) from the surface to the bottom of the OML and divide by the OML depth. If the OML depth were not constant in time and space, terms containing the derivatives of the OML depth should appear, but those terms are missing in Eq. (3.4). Please check the appendix of Moissan and Niiler (1998), for example.**
**Moisan, J. R., and P. P. Niiler (1998), The seasonal heat budget of the North Pacific: Net heat flux and heat storage rates (1950-1990), J. Phys. Oceanogr., 28, 401-421.**

Thank you so much for the reviewer's careful checking our OFGF. Before the estimation of all OFGF terms, we calculated the OML-mean quantities like,

$$A_{oml} = \frac{1}{D} \int_{D}^{surface} A \cdot dz$$

, here $A$ is arbitrary variable (temperature and currents) and $D$ is the OML depth. Therefore, our OFGF includes the changes in the OML depth itself implicitly. However, we did not sub-divided each partially-differeciation term like,

$$\left( \frac{\partial \theta_{oml}}{\partial y} \right) = \frac{1}{D} \frac{\partial \int_{D}^{surface} \theta}{\partial y} + \int_{D}^{surface} \theta \left( \frac{\partial (1/D)}{\partial y} \right)$$

in order to keep our discussions more simplified like, for example, Tozuka and Cronin (2014, *GRL*). The spatial-temporal changes in the OML depth also appear as the entrainment velocity according to Moissan and Niiler (1998) and we considered this term as residual term (because the accurate estimation of entrainment velocity is difficult from CFSR outputs). But, we did not mention this point explicitly in the previous manuscript. Now, we added more careful explanation on our OFGF and its

simplification in this study in the revised manuscript. Please see lines 152-178.

**2) Equation (3.5): The assumption that there is "no penetration of shortwave radiation beyond the OML to deeper ocean layers" may not be a good assumption considering that the OML depth could be as shallow as 20 m in the region during austral summer.**

Yes, this is a rough approximation. On the other hand, by even such approximation, the surface heat flux does not play a vital role for the frontogenesis of the ABFZ. Therefore, we could think that this rough assumption does not make large difference in terms of frontogenesis in this region. And probably, the frontogenetic effects of solar radiation are much less important than the longwave surface fluxes reslting from the combination of the SST front and cloud cover.

**3) How is the OML depth determined? Is it based on some density criterion? Also, how is delta theta calculated.**

We used the output of CFSR for the OML depth. The OML depth of CFSR is defined based on K-profile parametrization. We added this explanation. Please see lines 176. Delta(Theta) is estimated as the difference between the OML-mean value and the value at one-below layer of the OML. We had missed this explanation, and now it is added in the revised manuscript. Please see lines 167-169.

**4) Lines 389-392: Since the authors are using a reanalysis product, effects of data assimilation are also included in the residual term.**

We agree with it. We added more discussion on the data assimilation in the final section. Please see lines 493-495.

**5) Figure 5b: The tendency term should also be shown.**

Thank you so much for the suggestion. We added the tendency of ABFZ front in Fig.5a by green-colored line. Please note that the tendency is filtered through moving 30days filter. We added a description of the tendency in the revised manuscript. Please see lines 289-293 and new Fig.5a.

**Minor Comments**
**1) Figure 1: Label the left panel as Fig. 1a and the right panel as Fig. 1b (Line 169).**

Thank you so much for the comment. We added the labels to Fig.1.

**2) Although the editor provided many comments on their writing, there are still some grammatical errors etc.**

**Line 10: Replace "form" with "from".**

Corrected.

**Line 11: Delete ", respectively".**

Deleted.

**Line 30: Replace "Resason" with "Reason".**

Corrected.

**Line 36: Delete "over the southern African Continent" as this phrase is redundant.**

Deleted.

**Line 38: Replace "model" with "mode".**

Corrected.

**Line 49: Replace "GCM" with "CGCMs".**

Corrected.

**Line 57: Replace "insightfully" with "quantitatively".**

Corrected.

**Line 62: Replace "at" with "in".**

Corrected.

**Line 70: Replace "opposite " "opposing".**

Corrected.

**Line 72: Replace "more" with"most".**

Corrected.

**Line 86: Replace "put" with "make".**

Corrected.

**Lines 92-93: Replace "National Ocean and Atmosphere Association" with "National Oceanic and Atmospheric Administration".**

Corrected.

**Line 119: Replace "current velocity" with "zonal, meridional, and vertical current velocity,respectively,"**

Corrected.

**Line 129: Add "of" after "exchange".**

Added.

**Line 138: Replace "3,4" with "3.4".**

Corrected.

**Line 143: Replace "velocity," with "velocity and".**

Corrected.

**Line 144: Add ", respectively" after "depth".**

Added.

**Line 158: Replace "reminder" with "remainder".**

Corrected.

**Line 161: Replace "29-years" with "29-year".**

Corrected.

**Line 172: Delete "a" before "weak".**

Deleted.

**Line 176: Replace "shows" with "showed".**

Corrected.

**Line 193: Replace "exceed" with "exceeding".**

Corrected.

**Line 227: Replace "One" with "On".**

Corrected.

---

## Author Comment (AC2) · 5 Dec 2018

Reply to Reviewer#2

As said in the papers introduction, a quantitative analysis of the frontogenesis in the ABFZ is really not done yet. So, the paper deserves interest and may enhance our knowledge on the dynamics of the boundary between tropical and subtropical waters in Eastern Boundary Currents of the southern Atlantic. So I recommend to publish it.

However, I propose to revise the paper first, for two reasons. It has capacity to gain better scientific quality at some points and some steps in the reasoning are either not well described or not justified. The main crititics are on the way how the "residual" term is treated. Let us go through the paper in detail. Remarks are made in the "order of appearance". This mixes important and less important ones.

We greatly appreciate the reviewer for his/her quite constructive and helpful comments on our manuscript. As below, we have replied to all the comments and added more figure, sentences, and descriptions in the revised manuscript. Please note that the corrections in the revised manuscript are shown by **blue-colored font**. Please note that we changed the title of this manuscript to "**Frontogenesis of the Angola-Benguela Frontal Zone**" replying to the reviewer's comments. Also, please note that the number of equation in section 3 decreases from 5 to 4 following revisions.

**40) Does the SST and the ABFZ really influence the rainfall activity or are anomalies of both related to the same process?**

Here, we meant that the anomalies of SST in the ABFZ influence the rainfall activity regionally referring to Rouault et al., 2003 and Lutz et al., 2015. Both papers argue the moisture flux anomaly associated with the SST variability. Therefore, we added this explanation. Please see lines 41-42.

52) To reduce the SST bias in model results, a more realistic wind forcing is needed. This would reduce the overestimated poleward transport of tropical water. Understanding the ABFZ dynamics only does not reduce the model bias. But for sure, understanding something helps to deal better with it. It was a major outcome of the PREFACE projects that the SST bias is mostly due to shortcomings on the atmosphere side.

As we concluded, our approach of the OFGF could be a good diagnostic tool to evaluate the modeled ABFZ where most of CGCMs fail to simulate realistic SST. We agree that the wind forcing is important for the representation of the ABFZ (e.g., Koseki et al., 2018). An erroneous wind forcing leads to the bias of the ocean current and upwelling in the region. These oceanic currents and upwelling are important part of our OFGF. That is why we wrote that "understanding the ABFZ dynamics is to reduce the bias". However, as the reviewer mentions, unveiling the ABFZ dynamics does not contribute directly to reduce the bias of CGCMs. Therefore, we diluted the expression there. Please see line 54.

56) I think, Mohrholz et al. 1999, already gave some insight in the frontal "dynamics" in that sense that processes moving the front are identified. But you are right, a quantitative discussion in terms of hydrodynamic equations

**was not given there.**

Yes, we agree. However, Morholz et al. (1999) studies a case study of the ABZF in 1999 and the behavior of the ABFZ should include some inter-annual variability. In our study we focus more on the climatology of the ABFZ and its frontogenesis analyzing a long-term-mean seasonal cycle. Morholz et al. (1999) has been added as a reference. Please see lines 58-59.

71) Just to identify the "current systems", "Benguela current" and "Angola current", is not a discussion of the dynamics. The Benguela current has a more complex structure, one part that should contribute to the frontal dynamics, is the wind driven coastal jet. For the Angola current coastally trapped waves are important.

Our confluence term refers to the term expressed by  $(\frac{\partial \theta}{\partial y})(\frac{\partial v}{\partial y})$  and the Angola and Benguela Currents are southward and northward current. Therefore, we meant those currents can contribute to the frontogenesis of the ABFZ. We modified the expression on it. Please see lines 73-74.

**73) What is meant here? How does the specific cloud cover modify the heat flux components? Is it reducing or enhancing influence?**

Here, we considered that the low-level cloud formation modified the radiation budget at the ocean surface. If there is some meridional gradient in the cloud formation, shorwave and longwave radiation forcing can also have some gradient and it might affect the frontogenesis. We modified the expression there. Please see lines 75-76.

**99) .5 deg resolution is not really good? There are other ocean model products available with better horizontal resolution, e.g. from the ECCO group. What is the motivation of your choice?**

Thank you so much for raising quite a good point. Yes, some products from forced oceanic models have finer resolution. However, fine-scale wind forcing (based on satellite data) used in these models are only available for short periods whereas the .aim of our study is to provide a "climatological" view of the ABFZ. Moreover, since the satellite (scatermeter) winds cannot be observed in a coastal band of approximately 25-50km, the wind products used to forced high-resolution oceanic models likely has deficiencies in our region of interest. Therefore CFSR, which is based on a coupled ocean-atmosphere system is more appropriate for our study. We added this justification. Please see lines 104-111.

102) This justification for using the potential temperature does not hit the point. Please look into some textbook (say Olbers et al., Ocean Dynamics) for the in-situ tempera- ture transport equations. It is a highly complex one. The corresponding analysis of the frontogenetic terms would be a nightmare. Potential temperature or conservative temperature do not stand for a the thermodynamic quantity "temperature". But you are right, potential temperature at the surface is numerically equal to the in-situ temperature. Hence, potential temperature is suitable for your analysis and results can be presented as "temperature". Moreover, the approximate transport equation looks like a classical advection diffusion equation. May be you just say that you are considering potential temperature? Thats it.

Thank you so much for helpful comments. We changed "temperature" to "potential temperature" and removed the justification.

**119) What is the relation between Lagrangian and Eularian differentiations? As far as I see just commutation and product rule of differential calculus are considered.**

As the reviewer says, We use this relationship between Lagrangian and Eulerian,

$$\frac{d}{dt} \left( \frac{\partial \theta}{\partial y} \right) = \frac{\partial}{\partial y} \left( \frac{d\theta}{dt} \right) - \frac{\partial \vec{U}}{\partial y} \vec{\nabla} \theta$$

however, that description is a bit confusing and we removed it.

121) In the beginning, you made the very basic assumption that the front orients zon- ally. Now you consider the transformation of zonal gradients into meridional gradients. Do you consider zonal temperature gradients or do you neglect them? You basic assumption needs a better justification or zonal components of the fronts should be included? My view is that a major contribution to frontogenesis is upwelling and equatorward transport of cold water with the coastal jet. So zonal temperature gradients are very important. See the images in Mohrholz et al. 1999!

Thank you so much for a good discussible comment. This point is one of benefits of the frontogenetic function. Each horizontal component of the gradient of temperature is also generated by the other component through the shear effect. In the south of the ABFZ, the cool SST due to the coastal Benguela upwelling creates the strong zonal gradient of SST. The shear term could convert this zonal gradient to the meridional gradient and consequently, the ABFZ (defined as merdional gradient in this study) could be generated/maintained. We added these explanations in the manuscript. Please see lines 138-141.

**131) The equation is not correct. The vertical divergence of the solar radiation is missing.**

Yes, in this study, we assumed that the radiation is processed only in the ocean mixing layer, not penetrating below the ocean mixing layer. In addition, CFSR ocean data only has net heat flux data and therefore, we used the approximation. Since the net heat flux effect is secondary or negligible in the frontogenesis of the ABFZ, this approximation does not make large difference. This assumption was given in the original manuscript. Please see lines 173-175. In fact, the solar radiation component, which is included into Qs of the surface term SFLX of equation (3.4) in the text, should have been affected of the transmission coefficient (I(0)-I(D)). But, in this study, the coefficient is kept to 1 (all solar radiation is absorbed in the mixed-layer).

141) What do you mean with "production remains linear relation". I guess you mean that as long as the temperature is constant within the OML, the vertical averaging over the horizontal velocity can be carried out. Vertical mixing is not "replaced" by the heat fluxes, it is the upper boundary condition of the integrated mixing term. Q\_b is the lower boundary condition for the vertical heat flux.

Yes, that is what we meant. The previous expression was not clear, so we removed that sentence, "because the production remains linear relation". For the heat flux and vertical mixing, yes, vertical mixing and heat flux should be separated. We modified

Eq.3.3 like
$$\frac{\partial}{\partial y} \left( \frac{d\theta}{dt} \right) = \frac{\partial}{\partial y} \left( -\frac{\partial \overline{w'\theta'}}{\partial z} \right)$$
 with
 $\left( \overline{w'\theta'} \right)_{(z=0)} = SurfaceTurbulentheatflux$
 $\left( \overline{w'\theta'} \right)_{(z=D)} = Q_b$

and the explanation was also modified. Please see line 146-147 and 171-172.

**150) Please explain in which sense the mixing at the bottom is "higher order". In Fig. 4 the amplitude of the residual term is the largest of all contributions.**

We meant that the vertical mixing at the bottom of the OML is parameterized process in the ocean model of CFSR including non-linear processes and it may be difficult to make an accurate estimation of the vertical mixing with CFSR output only. Therefore, we decided to put the vertical mixing term to RESD. According to Giordani and Caniaux (2014), the vertical mixing is a large frontolytic term in the long-term period and our RESD also shows the frontolytic effect around the ABFZ. We understand that this treatment is a shortcoming of this study and will be desirable to address it with higher resolution ocean model (the output of temperature tendency due to the vertical mixing could be obtained in ocean model). We modified the final conclusion with addition of these shortcomings. Please see lines 480-486.

The vertical turbulent mixing term at the mixed-layer base Qb is represented according a K-profile parameterization in models (added this description, please see lines 175-178). As consequence an explicit expression of Qb is not possible from reanalysis unless we have access to the vertical profile of turbulent fluxes that is not usually the case from the reanalyses. However by using an expression of the exchange coefficient based on the Richardson number, it could be possible to derive an estimation of Qb but certainly not similar from that of reanalysis.

**161) This needs to be clarified. Which climatology? Do you calculate velocity and temperature climatologies first and from this the different contributions? Gaps in the model results? How can this be?**

First, we estimated each term of the OFGF for each year of 1979-2010 from dailymean data of CFSR. Then, we calculated the daily-climatology of each term of the OFGF. The calculation of climatology is the final step of estimation. We provided more descriptions of data processing in the supplemental information. Please see the supplemental file.

**170) I would propose to make clear somewhere that "strength of the ABFZ" or "intensity of the ABFZ" means the maximum value of the meridional SST-gradient, doesn't it?**

Thank you so much for the suggestion. We added a sentence to mention the

definition of the intensity of the ABFZ in this study. Please see lines 201-203.

187) Here it must be said that "dynamics" means analysis in terms of forces and responses. This is not done is this paper. Considering time changes or trajectories without considering the reasons, i.e., the forces, I would speak about "kinematics". In oceanography is becomes colloquial to say "dynamics" to address "time development". This is not correct in the original meaning of the word.

Thank you so much for correcting the word. Since this study focuses on the "frontogenesis" of the ABFZ with the OFGF, the frontogenesis is rather adequate than "dynamics". Therefore, we changed the title of this section to "Diagnosis on the frontogenesis of the ABFZ" (please see 220) and the title of this paper is to "Frontogenesis of the Angola-Benguela Frontal Zone".

**196) Today this cell is mostly called "Kunene upwelling cell", which is related to a very persistent wind patch off the mouth of river Kunene. Please check this.**

Thank you so much for it. We checked it and modified the name of cell with a new citation. Please see lines 229.

**225) This is not RESD of eq. (3.5). Please explain better.**

Here, our explanation was poor. In the section 5.1, we addressed the annual-mean climatology, which is independent of time. That is, the left-hand side of Equation 3.5 is zero (tendency in the SST gradient is zero). This zero is balanced with all terms on the right-hand side of Equation 3.5 (in the revised manuscript, it is Eq.3.4). Therefore, annual-mean climatology of RESD can be estimated in Eq (5.1). We added more explanation of annual-mean climatology RESD. Please see lines 256-262.

226) If you calculate the time average, the result is independent of time. Your reasoning is not convincing. Please describe, what you are really doing and how the quantities displayed in the figures are really calculated. Why don't you just calculate the residual term in 3.5? If you integrate the left hand side of 3.5 in time, you get the difference of the meridional temperature gradient between end and beginning of your time series. Is this difference really zero or small? You said earlier that the interannual variability is much higher than the seasonal variability. So, please add a figure that this difference is smaller than all the other terms. If this is not the case, your RESD is wrong.

As we replied above, section 5.1 is about annual-mean climatology and climatological RESD can be estimated because the tendency of the SST gradient (left-hand side of Equation 3.4) is zero. Our reasoning is not right as the reviewer argues. We removed the reasoning and added more explanations on how RESD is estimated from Equation 3.4. Please see line 256-262. Regarding that seasonal variability is smaller than the interannual variability, this is about the location of the ABFZ between "climatological seasonal-scale" and inter-annual variability shown in Fig. 3. We added "climatological" before seasonal scale. Please see line 204.

Actually, in section 5.1, we just addressed time-independent feature of the ABFZ and OFGF (annual-mean climatology) and section 5.2 we handled the climatological

seasonal cycle of the ABFZ and OFGF where there is a tendency of the SST gradient and the way of estimation of RESD is also different from that in section 5.1.

**236) As said before, TILT and CONF are kinematic terms. So they do not "drive" the generation of the front. Something drives the currents causing tilt and confluence. This is mostly the wind (local and remote) and vertical heat fluxes. But for sure TILT and CONF are the main contributions to the OFGF.**

Thank you so much for the helpful comment. We changed the word of "driver" to "source". Please see line 271.

**238) The phase shift is an interesting point and needs more attention.**

Thank you so much for the note. We added more description of the phase shift and seasonal cycle in the revised manuscript. Please see lines 273-274.

242) The box integral stands for the time derivative of the north-south temperature difference. This could be, one may assume this, a constant, but the temperature gradient inside the box may vary between a linear one to a sharp front inside the box. Is the average over the box a good measure for the overall strength of the ABFZ? The maximum gradient, a median of the gradient or the meridional temperature variance may be also an interesting quantity.

We agree, as the reviewer mentions, there could be some variability in the ABFZ in the averaging box. On the other hand, the meridional location of the ABFZ is almost stable in the climatological seasonal scale as shown in Fig.2 and we would think that the box can capture the representative features of the ABFZ since the box covers the seasonal maximum of the ABFZ. We added this justification in the revised manuscript. Please see lines 280-282. Yes, such quantities like median and variance are also interesting quantities to be addressed, however, this study, in the first place, focuses on more general feature of the ABFZ. Thank you so much for the suggestion.

**259) Please do not mix variables and units. I propose to introduce \Delta t and say this quantity is 1d.**

Thank you for the comment. We modified the equation and provided a bit more explanations. Please see lines 301-303.

281) The "two different physical processes" are a very important hypothesis. I think that the paper of Mohrholz et al. (1999) shows the action of TILT and CONF in a descriptive manner. In the new paper we see the equations.

Thank you so much for the comment. We added Mohrholz et al. (1999) as a reference there. Please see line 323.

**300) I cannot see the "mirror image".**

This expression is too exaggerating. We removed it.

**302) To see this, a graticule or some "zero line" should be added in Fig. 5 and 6.**

Thank you so much for the suggestion. We added the zero lines to Figs. 5 and 6.

309) Which SSH? From the model? How did you calculate OML-depth? Why only the geostrophic component? Some more details in the "data"-section would be of great help. More important, the concept of reasoning should be made clear. So, what is the motivation, to consider geostrophy only? Fig.7 does not show the Ekman transport, which should be an essential part of the surface flow in the trade wind zone. I would expect it as divergent from the pronounced wind stress curl. If not, the model would be fully Sverdrup balanced away from the coast? Please add some remarks on your way of reasoning and comment.

Yes, we used SSH data from CFSR. But as we had not mentioned it in the previous version of the manuscript, we now added some descriptions in the data section. Please see line 113-114. The reason why we focused on the geostrophic current is because the lower SSH, the Angola Dome (e.g., Doi et al., 2007), forms around the ABFZ exhibiting a pronounced seasonal cycle. This variation in SSH can induce the variability in the geostrophic current and therefore we looked at the geostrophic current does not perfectly explain the all variability in CONF (Fig.7), but its large part contributes to the CONF and its variability also shown in Fig.7. The difference between  $V_{oml}$  and  $V_g$  can be a component due to Ekman transport and other ageostrophic processes. We added the reason why the geostrophic component is surveyed and this discussion. Please see lines 349-354 and lines 370-374.

336) A geostrophic balance between a pressure gradient and the Coriolis force of a current does not imply any information on the driving mechanisms. The Angola Gyre may be related to the wind stress curl in that area but the doming structure and the currents develop together. For me it seems to be more reasonable that the dome is result of a flow, but for sure not of the geostrophically balanced one. It is not intension of this paper to explain the dynamics of the dome. Just, please avoid oversimplified unphysical wording. Proposal: replace "to induce" by "related to".

Thank you so much for the suggestion. Yes, the dynamics of the Angola Dome is out of scope of this study and already done by Doi et al. (2007). In this study, we make a bridge the seasonal variability in ocean current and the Angola Dome. We replaced "induce" to "be related to". Please see line 387.

349) what is negative stratification? In Eqs. 3.4 and 3.5 the sign convention of \Delta \theta is not explained. I tried to understand 349/350. Does it mean negative dw/dy and negative stratification (whatever this is) from January to August and positive dw/dy and positive stratification otherwise? Also 354 to 357 leaves me confused. Aug-Sep. dw/dy is negative but is positive from August to December. Not quite clear. What is Fig. S1a/b? It cannot be Fig. 1 a/b since this does not show w.

At line 349, our explanation was confusing. Now, we changed it to "the negative dw/dy and the positive stratification". The negative dw/dy is that the upward motion becomes weaker at lower latitude and the positive stratification means that the OML-mean temperature is higher than the temperature at the bottom of the OML. Basically, the stratification is always positive, on the other hand, the dw/dy becomes positive in occasion (especially in August-September as shown in Fig. 8). According to Eq 3.4, the negative dw/dy and positive stratification generate the frontogenesis and the positive dw/dy creates the frontolysis. Figures S1a and b give the horizontal distribution of dw/dy in March-April and August-September, respectively. We

modified some sentences in this paragraph. Please see lines 399-404.

**360) What about wind driven mixing?**

Yes, that contribution is considerable for the determination of the OML depth. However, we used the OML depth that has been already calculated and archived in CFSR data. Therefore, we cannot separate the thermodynamical and dynamical components for the OML depth.

**367) Where are the Figures?**

We gave Figs. S1c and d as supplemental information.

372) This was a chapter on vertical velocity. So, I would have upwelling in my mind, but the chapter closes without even mentioning "wind". So, as a discussion, it remains very technical and less physical. The discussion is mostly on the kinematics of the front and not on the dynamics (in terms of forces).

We agree that we did not show the surface wind in the previous version of the manuscript. Now, we added Figure S2 that shows the annual-mean zonal and meridional wind stress and the Ekman divergence for CFSR and SCOW estimated by the wind stress yielded by

 $M_x = \frac{\tau_y}{f}$ ,  $M_y = -\frac{\tau_x}{f}$ , here *M* is Ekman transport and  $\tau$  is wind stress. The Each component of the Ekman divergence is also estimated by partial differenciation like  $\frac{\partial M_x}{\partial x}$  and  $\frac{\partial M_y}{\partial y}$ . The Ekman divergence induces the vertical velocity in the Ekman layer. Please note that  $\frac{\partial M_y}{\partial y}$  is shown in Fig.S2a and c and  $\frac{\partial M_x}{\partial x}$  is shown in FigS2b and d. This figure evidences that CFSR is able to capture the climatological wind stress around the ABFZ (although there are some differences from the satellite obsrvation). Actually, the zonal and meridional components of the Ekman divergence are comparable around the ABFZ. Interestingly, the zonal wind stress curl is also responsible for the upwelling around the ABFZ (CFSR and SCOW show in Fig. S2a and c). We had thought that this result was included in the original manuscript, however, the vertical velocity at the bottom of the OML ( $w_b$  in Eq. 3.4) is not completely consistent with the vertical velocity due to the Ekman divergence (partially because the OML and Ekman layer is not consistent perfectly). Therefore, we decided not to investigate deeply in this study. On the other hand, we added some description on wind stress with Fig.S2 in Section 7. Please see lines 438-441.

**380) higher order with respect to which parameter?**

As we replied above, this is due to the complexity of parameterization of the vertical mixing. We modified the expression there. Please see lines 430-432.

**381) Do you mean "vertical mixing at the bottom of the OML"?**

Yes. We added "at the bottom of the OML". Please see line 431.

**391/2) two maxima of what? Maximum gradient?**

Yes. We added "of the SST meridional gradient". Please see line 445.

414) The variability of the frontal position is not discussed in this paper. However, the common view on the ABFZ variability is that of a seasonally varying meridional movement of the fronts. The stripe between  $17^{\circ}S$  and  $15^{\circ}S$  is relatively narrow. Please show that you cover the major part of the ABFZ during all model years.

Yes, we agree that there is an inter-annual variability in intensity and location (in particular, meridional direction) of the ABFZ (e.g., Rouault et al.,2017; Mohrholz et al., 1999). On the other hand, in this study, we focus on the "climatological" features of the ABFZ (we emphasized "climatology". Please see line 82). In the climatological seasonal cycle, as Fig.2 shows, the location of the ABFZ is almost stable and the band 17S and 15S covers the maximum intensity of the ABFZ. Therefore, we used the relatively narrow box. However, as the discussion on the inter-annual variability is also important for the further investigation, we added more discussion on the variability in Section 7. Please see 480-486.

**Fig. 2) Averaging from 10°E to 12°E excludes the coastal areas north and south**

**of 17°S. Fig. 4 shows that this may be a poor approximation. The same applies for other figures using the same approximation. I propose to select a stripe along the coast, probably with a width of some Rossby radii?**

Thank you so much for suggestion. When we made the plots of Fig.2, we confirmed that the coastal grids are included in the calculation. In all of our figures with averaging, the coastal grids are included in the calculations. Since CFSR has only  $0.5^{\circ} \times 0.5^{\circ}$  resolution, the coastal area is not well resolved. Therefore, the investigation with higher-resolution ocean model will be desirable. We added this discussion briefly. Please see lines 487-495.

Fig. 4) I propose to add a figure showing the time dependent term. It should become small for a long averaging period. But this should be shown. Somewhere in the text the strong interseasonal variability was mentioned. So it could be that this term is large. If the interannual variability is larger than the seasonal cycle, how significant is this figure? For me it is not really satisfying to see that RESD is the largest contribution. I would suggest to write down the details on how the averaged terms (as well as the monthly climatologies) are calculated from the daily model output. I think the only term, which cannot be calculated is that related to vertical diffusion at the base of the OML. Even here at least the order of magnitude could be estimated.

Thank you so much for suggestion. As we replied above, Fig.4 is the annual-mean climatology and therefore, the tendency in the SST meridional gradient is zero and RESD has been calculated from Eq3.5 (please note that Eq3.4 in the revised manuscript) in letting the left-hand side zero. On the other hand, in Section 5.2 (seasonal cycle), we are showing the results of the climatological seasonal cycle and added a plot of the tendency to the SST meridional gradient in Fig.5a. According to Giordani and Caniaux (2014), the vertical mixing is large frontolytic source and our RESD includes the vertical mixing and strong frontolysis. We added the post-processing how to calculate the climatology of the OFGF terms. This additional information is given in supplemental information.

Fig 5b) I am missing the other contributions. A zero line would be of some help. The same applies for the following figures.

We added SHER and SFLX terms in Fig5b. Zero lines for figures of time series were also added.

In summary the paper is of great interest. The message is that just "normal" linear advection and mixing even with coarse resolution may generate a frontal system. However, the word "dynamics" is to promising. The paper considers kinematiks of the fronts, the consideration of dynamics, i.e., a discussion of the relation of the development of the front in relation to atmospheric or other driving forces is still outstanding. Unfortunately the typical wind pattern are neither mentioned nor included in the discussion. However, we should not expect that the authors do in one paper what a whole community could not accomplish within decades. So my proposal would be, not to include an analysis of the winds here and just to consider the paper as a step into this direction.

Thank you so much for very encouraging comments. Regarding the word of dynamics, we diluted the word in the revised manuscript, and changed the title of this paper to "Frontogenesis of the Angola-Benguela Frontal Zone" as we replied above. Also, we added Fig.S2 showing the annual-mean climatology of the wind stress and Ekman divergence associated with the wind stress. This figure can be an implication that the vertical velocity is also induced by the zonal wind stress curl.

More details on the data processing are needed, especially on the time averaging to gain the climatology. Sometimes I suspected that contributions to the OFGF are calculated from climatological quantities. Please make a clear statement on the method.

As we replied, we added the explanation how to process the CFSR data to calculate the climatology. Please see supplemental information.

The residual term is to large to be called residual. The remaining terms collected here should get a name and deserve more detailed discussion. What about a \theta/D dw/dx term? The reasoning that the left hand side of Eq. 3.5 is small, which defines RESD (Eq. 5.1) is not convincing. This problem should be reconsidered, Eq. 5.1, if it holds, requires more justification.

As we replied, our RESD term includes the vertical mixing at the bottom of the OML. This term may be a large part of RESD. Because the calculation of RESD is based on residual of Eq.3.5 (please note Eq3.4 in the revised manuscript), it might be better to keep the name of RESD even though its contribution is large. Therefore, we added more discussion on RESD and vertical mixing in Section 7. Please see lines 487-495.

Also, "residual" does not mean necessarily that RESD is small. "residual" refers to processes which were not directly computed from the reanalysis data. These processes were estimated from the closure of equation (3.4) because this is the best way to minimize errors which affect turbulence and advection terms.

The fact that the confluence of the geostrophic flow is the major contribution

to the CONF-term is important and should be discussed more explicitly. Fig. 7C shows a strong divergence of the geostrophic flow. The water must stay somewhere. Here it continues most probably as the zonal component of the Ekman transport. Since this term is not in the OFGF, it does not become visible here. So one may conclude that a consideration of geostrophic currents may be sufficient to understand the time evolution of the frontal system. I am not convinced that this is true. Please consider this issue.

Based on our analysis, the confluence around the ABFZ is approximately estimated by the geostrophic current. However, as we replied above, there are some differences between geostrophic current and OML-mean current. These differences can be due to the Ekman transport and ageostrophic component. We added this discussion in the revised manuscript. Please see lines 370-374 and 466-468

---

## Author Response (AR2)

**Reply to the editor**

We would like to appreciate the editor for his constructive comments on our manuscript after his carefully checking. We corrected and modified our manuscript following his comments. Also, we read carefully the manuscript again and made some cosmetic corrections of the manuscript. Please note that all the modified parts are shown by red-color font in the revised manuscript.

**Comments**
**Line 51. "CGCMs"?**

Corrected.

**Line 110. ". . available for only a relatively short time, limiting . ."?**

Corrected.

**Line 121. ". . is plentiful literature . ."?**

Corrected.

**Line 122. ". . referred to." Or " . . referenced."**

Corrected.

**Line 123 delete final ",". Line 125 delete ","**

Corrected.

**Line 130. "and using"**

Corrected.

**Line 158. Delete first ",". ". . with subscript oml . ."**

Corrected.

**Line 161 (Ref 2 commented). ". . production is linear in u, v, as long"?**

Corrected.

**Lines 168-169. ". . temperature just below the OML . ."?**

Corrected.

**Lines 182-188 (resuming Editor comments). I think the "new" and former text could be integrated better. As I understand it (i) entrainment velocity at the bottom of the OML, (ii) Qb and (iii) advection of $\partial\theta_{oml}/\partial y$ are all in RESD but (i) is small. (iii) is not frontogenetic (I agree) but it could cause incorrect diagnosis of frontogenesis in the Eulerian view.**

Thank you so much for the comment. We combined the new and previous texts for RESD. Please see lines 183-189.

**Lines 192-195. The main text needs to refer to the supplement here (c.f. Ref 2 comment "161" and later). In relation to this and Ref 2 comment "242", by analogy with time-averaging stratification, it occurred to me that for the climatology sharp gradients would be retained better by time-averaging y for given temperature (I assume you time-averaged temperature for given location). When there is (inter-annual) variability, time-averaging for a given location spreads out the temperature distribution. This bears on section 4 et seq.**

We added referring the supplemental information. Please see lines 191-192.

**Lines 257-261. This is clumsy with the additions to the text. Better ". . RESD is estimated from (3.4) where the left hand side $\partial/\partial t(\partial\theta oml/\partial y)$ is zero for climatology independent of time: RESD = . . . (5.1).**
**Note that all terms . ."**

Corrected.

**Line 264. "at least" is unclear. "at least in part" or omit?**

Corrected.

**Line 278. "temporal" -> "time"**

Corrected.

**Line 284. Omit "respectively" (twice)**

Corrected.

**Line 301 (Ref 2 commented). I think θoml is a function of (respectively) t, t+Δt, t-Δt) and the denominator on the right-hand side should be 2Δt. (I guess then Δt = 1 day). You would need to change line 302 accordingly.**

Corrected.

**Lines 350, 377&379. It is strange to see a (Angola) Dome being identified with low SSH. I am not familiar with the area and associated names but some explanation would help assure similarly unfamiliar readers that there is not a mistake here.**

The term "dome" is reffered to the vertical structure of temperature in this area that evokes a cold "dome".The first definition of Angola Dome is my Mazeika (1976) as cold dome. We added this citation and brief explantaion of Angola Dome. Please see lines 349-350.

**Line 385. Better omit "up to"**

Corrected.

**Line 399. ". . points out the negative $\partial wb/\partial y$ and the positive stratification . ." to correspond with your response to the referee and the figure?**

Yes. We corrected.

**Line 410. "forcing and wind-driven mixing. Also . ." in accord with Referee 2 comment "360" and your response.**

Corrected.

**Line 438. Omit second "the"?"**

Omitted.

**Reply to the reviewer#1**

We would like to appreciate the reviewer#1 for his/her constructive comments on our manuscript We corrected and modified our manuscript following his/her comments. Please note that the modified parts are shown by red-color font in the revised manuscript.

**Lines 183-184. (Referee 1 comment). I generally agree with the authors regarding their response to my Major Comment #1, but I do not think entrainment is a higher order term. Thus, I recommend the authors to replace "its contribution is of higher order and it might be" with "it is".**

Corrected.

---

## Author Response (AR3)

To the Editor of Ocean Science
Dear Prof. John M. Huthnance,

We would like to upload our corrected manuscript entitled, "Frontogenesis of the Angola-Benguela Frontal Zone" written by Shunya Koseki (corresponding author), Herivé Giordani and Katerina Goubanova.

In this correction, we have corrected our manuscript point-by-point following the editor's comments. In this submission, we upload the corrected manuscript, figures, text file, and supplemental information individually.

We really appreciate the editor for addressing our manuscript, kind consideration, and final decision on our manuscript.

Sincerely,
Shunya Koseki

Shunya.Koseki@gfi.uib.no
+47 55 58 98 24
Geophysical Institute,
University of Bergen,
Allegaten 70, Bergen, Norway